# Growth Factors and the Choroid Plexus: Their Role in Posthemorrhagic Hydrocephalus

**DOI:** 10.3390/biomedicines13061366

**Published:** 2025-06-03

**Authors:** Hong Ye, Wei Miao, Richard F. Keep, Jianming Xiang

**Affiliations:** 1Department of Neurosurgery, University of Michigan, Ann Arbor, MI 48109-2200, USA; yehong@kmmu.edu.cn (H.Y.); miaowei@kmmu.edu.cn (W.M.);; 2Department of Pathology, Kunming Medical University, Kunming 650500, China; 3Department of Neurology, The 2nd Affiliated Hospital of Kunming Medical University, Kunming 650101, China; 4Department of Physiology, University of Michigan, Ann Arbor, MI 48109-0617, USA

**Keywords:** choroid plexus, hemoglobin, intraventricular hemorrhage, posthemorrhagic hydrocephalus

## Abstract

**Background/Objectives:** Intraventricular hemorrhage (IVH) frequently occurs in premature infants and adults with intracerebral or subarachnoid hemorrhage. It is a major cause of cerebral palsy in premature infants and a risk factor for poor outcome in adult cerebral hemorrhage. Posthemorrhagic hydrocephalus is a common complication of IVH and aggravates brain damage. Hemoglobin (Hb), released from the hemorrhage after IVH, has been implicated in IVH-induced hydrocephalus. The aim of the current study was to examine the impact of Hb on the choroid plexuses (CPs) that reside in the ventricular system. **Methods:** Experiments were performed in freshly isolated CPs, in primary cultures of CP epithelial cells (CPECs), and in the Z310 cell line exposed to Hb with MTT assay, scratch wound healing assay, cell counting/total cell protein measurement and RT-qPCR. **Results:** We found that Hb significantly induced CPEC proliferation (e.g., 37–65% higher than control by MTT assay and 56% higher than control by cell counting), and upregulated mRNA expression of growth factors in isolated CP tissue (e.g., IGF-2 and NGF were 39% and 79% higher than control by RT-PCR). Hb also remarkably induced mRNA expression of NKCC1 (50%) and claudin-2 (154%), two proteins involved in CSF secretion, in isolated CP tissue. **Conclusions:** These results indicate that Hb-induced growth factor-mediated CP proliferation and upregulation of CSF secretion-related proteins might contribute to PHH and suggest there may be alternate therapeutic targets for PHH.

## 1. Introduction

Intraventricular hemorrhage (IVH) is a common occurrence after intracerebral hemorrhage (ICH) and subarachnoid hemorrhage (SAH) in adults [1,2] and germinal matrix hemorrhage in premature infants [3,4]. Posthemorrhagic hydrocephalus (PHH) is a serious complication of IVH in both adults and premature infants [3,4,5,6], with PHH being a risk factor for poor outcome and morbidity [2,7,8]. The underlying mechanisms of PHH are not fully understood. It may initially result from a block in CSF flow pathways due to the IVH, but there is also evidence that clot-derived factors, such as hemoglobin (Hb) and iron, can cause ventricle dilation [9]. In addition, there can be fibrosis at CSF outflow sites limiting CSF reabsorption and a number of studies have indicated the involvement of transforming growth factor b (TGFb) in hydrocephalus [10,11].

While most attention has focused on the role of alterations in fluid flow through and out of the CSF system in hydrocephalus (obstructive or communicating), CSF over-production may also occur [12,13,14,15,16]. The choroid plexuses (CPs), in the lateral, 3rd, and 4th ventricles, produce the majority of CSF via the vectorial transport of Na, Cl, and HCO_3_ from blood to ventricle across the CP epithelium. CP epithelial cells show a polarized distribution of many ion transporters [17,18,19,20]. For example, the Na/K/Cl cotransporter, NKCC1, is present on the apical membrane, where it may be involved in CSF secretion [21,22] although it has also been suggested that it can be involved in CSF reabsorption, reducing hydrocephalus [23]. They also express the tight junction protein, claudin-2 [24,25]. Unlike many claudins, it can form a fluid-permeable channel between epithelial cells. Stromal macrophages in CPs also play a role in iron-induced hydrocephalus, and mast cells in CPs dramatically increase during tumor-associated hydrocephalus.

As well as changes at the level of the individual CP epithelial cell, another potential determinant of overall CP function is CP size. CP tumors can cause CSF overproduction [26]. There is evidence that IVH can cause cell death in the CP epithelium in vivo and that this can be mimicked in vitro by met-Hb and heme [27]. An unexplored area is potential repair mechanisms at the CP epithelium that may limit the effects of such injury. CPs have become a target for a novel therapeutic strategy for hydrocephalus [28].

The aim of the current study was to examine the impact of Hb on CPs and whether CPs play a role in PHH. The current study examined CP response to Hb with a cell line, primary culture cells, and tissue isolated from animals.

## 2. Materials and Methods

This study was approved by the University of Michigan Committee on the Use and Care of Animals. Sprague Dawley rats used in this study were purchased from Charles River Laboratories (Chicago, IL, USA) and housed in a temperature-controlled room with a 12 h light/12 h dark cycle to mimic natural rhythms and had ad libitum access to water and food. No human tissues or cells were used in this study.

Cell culture medium, fetal bovine serum (FBS), growth factors, insulin and transferrin were purchased from Thermo Fisher Scientific (Grand Island, NY, USA) unless otherwise specified. Hydrocortisone and laminin were purchased from Sigma-Aldrich (St. Louis, MO, USA). Z310 cells (a rat immortalized CP epithelial cell line) were from Dr. Wei Zheng (Purdue University). SYBR Green PCR Mix was purchased from Bio-Rad Laboratories (Hercules, CA, USA).

### 2.1. Cell Culture

Primary CP epithelial cell (CPEC) cultures were derived from the lateral and 4th ventricle CPs of 1- to 2-day-old Sprague–Dawley rats as previously described [29]. Briefly, CP epithelial cells were isolated using CP digestion, sedimentation and centrifugation, followed by differential attachment on plastic. The collected epithelial cells were seeded on laminin-coated normal culture plates for MTT and scratch wound healing assays as well as PCR. The culture medium was changed every 48 h after seeding and consisted of Dulbecco’s minimum essential medium (DMEM)/F-12 (1:1) supplemented with 10% (*v*/*v*) fetal bovine serum, 2 mM glutamine, 25 mg/mL of gentamicin, 5 mg/mL of insulin, 5 mg/mL of transferrin, 5 ng/mL of sodium selenite, 10 ng/mL of epidermal growth factor, 2 mg/mL of hydrocortisone, and 5 ng/mL of basic fibroblast growth factor. Cells were grown in a sterile incubator at 37 °C, 95% relative humidity, and 5% CO_2_. Cultures were inspected visually for epithelial growth (i.e., cobblestone appearance) on a regular basis. Confluence occurred 7–10 days post-seeding. Z310 cells were cultured in DMEM with 10% FBS.

### 2.2. Isolated Rat Choroid Plexuses

Sprague Dawley rats (30–50 days old) were anesthetized with pentobarbital (100 mg/kg) and euthanized. The lateral and 4th ventricle CPs were dissected and incubated in culture medium in a sterile incubator at 37 °C, 95% relative humidity and 5% CO_2_ for 10 min before further experiments.

### 2.3. Hemoglobin Treatment

Hb was purchased from MP Biomedicals (cat# 151234, Irvine, CA, USA). Initially, it is a mixture of met- and oxyhemoglobin, but incubations were performed in a 20% oxygen atmosphere. The Hb was dissolved in culture medium without FBS and diluted to desired concentrations. Cells and isolated CPs were treated with or without 10 µM or 30 µM Hb in culture medium without FBS for 6 or 24 h. At the end of the treatment, cells and isolated CP were washed three times with PBS and then harvested for real time PCR assay.

### 2.4. MTT Assay

Cells were grown on 96-well plates and treated with or without Hb for 24 h before the MTT (3-(4,5-dimethylthiazol-2-yl)-2,5-diphenyltetrazolium bromide) assay. They were then washed three times with PBS before 10 µL of MTT reagent (ATCC, Manassas, VA, USA) in 100 µL of culture medium was added to each well and the cells incubated for 3 h. 100 µL of Detergent Reagent was added when the purple precipitate was clearly visible under the microscope. Plates were covered in the dark and stored at room temperature for 3 h and then read at 570 nm in a Bio-Tek microplate reader.

### 2.5. Scratch Wound Healing Assay

Cells were seeded in 24-well plate and a 28-gauge needle was used to scratch the surface of plate when cells reached about 80% confluence. After imaging the initial cell gap under a microscope, cells were treated with or without 10 µM or 30 µM Hb. The same areas were then re-imaged after 12 h and 24 h. ImageJ (version 1.5, NIH, Bethesda, MD, USA) was used to measure the width of the cell gap before and after Hb treatment, and the gap expressed as a % of initial.

### 2.6. Cell Counting and Total Cell Protein Measurement

Z310 cells were seeded in 12-well plates and incubated for 24 h. Then, the cells were treated with or without 30 µM Hb in culture medium without FBS for 24 h. At the end of the treatment, the medium was removed, and the cells were washed with PBS for three times. 0.5 mL of 0.25% trypsin was added to each well and incubated for 5 min. 0.5 mL of culture medium was added to each well and the cell suspension was well mixed with a 1000 µL pipette. A total of 2 × 10 µL of the cell suspension was used for cell counting with Invitrogen Countess. Then, the cell suspension was spun down, resuspended in PBS, and sonicated for total protein assay with Protein Assay Dye (Bio-Rad Laboratories, Hercules, CA, USA).

### 2.7. RNA Isolation and cDNA Synthesis

Total RNA was extracted from primary CP epithelial cells and isolated CP tissues using TRIzol Reagent (Life Technologies, Carlsbad, CA, USA). In brief, cell samples were homogenized in TRIzol Reagent and incubated for 5 min. Chloroform was added to each sample, incubated for 3 min, and then centrifuged at 13.3× *g* for 15 min at 4 °C. The upper phase was collected, isopropanol was added and incubated for 10 min. Samples were centrifuged (13.3× *g*) for 10 min at 4 °C and the supernatant was removed. Subsequently, RNA was washed with 75% ethanol, dissolved in RNase-free water and RNA concentrations were measured on Nanodrop 2000 (Thermo Fisher Scientific, Grand Island, NY, USA).

cDNA synthesis was performed using the cDNA Reverse Transcription Kit with RNase Inhibitor (Applied Biosystems, Waltham, MA, USA). Each sample contained 1 µg total RNA in a 20 µL reaction mixture, which was incubated at 25 °C for 10 min, 37 °C for 120 min, and 85 °C for 5 min.

### 2.8. RT-qPCR

RT-qPCR was performed to examine mRNA expression of targeted growth factors, CSF secretion-associated proteins and HO-1 (heme oxygenase-1) in primary CP epithelial cells, Z310 cells and isolated CP tissues after treatment with Hb or control. For each PCR reaction, 5 ng cDNA and 10 µM forward and reverse primers (Table 1) were mixed with SYBR Green PCR Mix in a reaction volume of 20 µL. The samples were incubated at 95 °C for 3 min, followed by 40 cycles of 95 °C for 15 s and 60 °C for 1 min, and analyzed in the Eppendorf Realplax^2^ Mastercycler (Epgradient) (Hauppauge, NY, USA). Each reaction was analyzed in triplicate. The quantification of target genes was normalized to GAPDH using the ΔΔCT method.

### 2.9. Statistical Analysis

The results are expressed as the mean ± SEM from at least three separate experiments. Data were analyzed using *t*-test or one-way analysis of variance (ANOVA), followed by Dunnett’s post hoc test for comparisons to a single control, or Tukey’s post hoc test for multiple comparisons between groups. A *p* value less than 0.05 was considered statistically significant (*, **, and *** indicate *p* < 0.05, *p* < 0.01 and *p* < 0.001, respectively). All analyses were performed with Prism (Version 10.4.2, GraphPad, San Diego, CA, USA).

## 3. Results

The effects of Hb on cell proliferation were examined by MTT and scratch wound healing assays. Exposure to 10 and 30 µM Hb for 24 h significantly increased cell growth in Z310 cells and primary rat CPECs, as assessed by MTT assay (Figure 1). Compared to controls, Z310 cell growth increased by 37 ± 5% and 65 ± 7% with 10 µM and 30 µM Hb, respectively (Figure 1A). Similarly, primary CPEC growth increased by 24 ± 4% and 74 ± 9% with 10 µM and 30 µM Hb, respectively, compared to controls (Figure 1B).

To confirm the results of the MTT assay indicating Hb-induced cell proliferation, cell numbers were counted and the total protein amount was measured after Z310 cells were exposed to the treatment with or without Hb (30 µM) for 24 h. Compared to control, the number of cells increased by 56 ± 22% (n = 5, *p* < 0.05) and the total cell protein amount increased by 27 ± 7% (n = 9, *p* < 0.005) with Hb treatment.

In the scratch assay, induced wounds at 12 h were 4–5-fold smaller in Z310 cells exposed to 10 and 30 µM Hb compared to control cells (Figure 2A,B). Similarly, with primary rat CPECs assessed at 24 h, induced wounds were 4.5 ± 1.1 and 4.0 ± 1.2 times smaller in 10 and 30 µM Hb-exposed cells compared to control cells (Figure 2C,D).

As expected, Hb exposure induced a stress response in CPEC and Z310 cells, with marked increases in the expression of heme oxygenase-1 (HO-1, also known as heat shock protein 32) (Figure 3). However, Hb might also affect growth via an autocrine response due to the release of growth factors from the CPECs. Compared to control, mRNA levels for brain-derived neurotrophic factor (BDNF), insulin-like growth factor (IGF)-2, transforming growth factor beta-1 (TGF-β1) and nerve growth factor (NGF) in isolated CP tissue were significantly increased (80 ± 6, 39 ± 12, 64 ± 20 and 78 ± 24%) after 6 h of treatment with 10 µM Hb (Figure 4). There were no significant differences in TGF-β2, TGF-β3 or basic fibroblast growth factor (bFGF)-2 mRNA levels (Table 2). It should be noted that the baseline expression of IGF-2 (relative to GAPDH) was much higher than the other growth factors examined (Table 2).

Exposing isolated CPs to 10 µM and 30 µM Hb for 6 h increased Na-K-Cl cotransporter 1 (NKCC1) mRNA expression by 39 ± 15% and 58 ± 13%, respectively, compared to control (Figure 5A). In contrast, there were no changes in Na/K-ATPase subunit expression (Table 2). Six-hour treatment with 10 and 30 µM Hb also increased mRNA expression of claudin-2 (a pore-forming tight junction protein) in isolated CP tissue (2.5 ± 0.5 and 3.8 ± 1.2 times higher than control; Figure 5B). As expected, exposure to Hb significantly increased expression of HO-1 in isolated CP (Figure 6).

## 4. Discussion

The impact of IVH on the CP and whether changes in the CP contribute to or limit PHH has received relatively little attention. Hemoglobin is a major component of an IVH that has been implicated in PHH. The current study found that: (1) Hb induces CPEC cell proliferation in both the Z310 cell line and primary cell cultures; (2) Hb induces wound healing in CPECs as assessed by a scratch assay; (3) The effects of Hb on proliferation and wound healing may be related to an induction of specific growth factor expression by CPECs; and (4) Hb induces expression of the Na/K/Cl cotransporter (NKCC1) and the pore-forming tight junction protein, claudin-2, which may be important in CSF secretion.

CSF is predominantly secreted by the CPs in the lateral, 3rd, and 4th ventricles. After traversing the ventricular system and entering the subarachnoid space, CSF is absorbed into the bloodstream directly or via the lymphatic system [30]. Any pathological conditions affecting its movement, absorption, or production may result in hydrocephalus [31,32]. Besides secreting CSF, the CPs also form the blood-CSF barrier (BCSFB) which helps maintain the extracellular milieu of the brain by actively modulating chemical exchange between the CSF and blood, surveying the chemical and immunological status of the brain, detoxifying the brain, secreting cytokines, and participating in brain repair processes following injury [33,34,35,36]. Despite the importance of the CPs and their proximity to intraventricular blood, the effects of IVH on CP have received relatively little attention.

Clot-derived factors, including Hb, iron, and thrombin, play a major role in hemorrhagic stroke-induced brain injury [37,38,39]. Examining individual clot components simplifies studying the effects on the CP in vitro. For example, a previous study found that IVH-induced CPEC death in vivo was mimicked by exposure to met-Hb or heme in vitro [27]. There are a number of in vitro preparations available for studying the CP in vitro including freshly isolated CPs, primary cultures of CPECs and CPEC cell lines, such as Z310. The current study uses all three types of preparation. The freshly isolated CPs have the advantage of most closely reflecting the in vivo CP. However, they have limited long-term viability and are not solely composed of CPECs. The primary CPEC cultures and Z310 can be used for longer experiments, are solely comprised of CPECs, and there is access to the apical and basolateral membranes (if grown on transwells). The immortalized Z310 cell line is the easiest to grow and maintain but furthest from the in vivo CPECs. In the current study, the effects of Hb on Z310 cells were mimicked by effects on CPEC primary cultures.

As noted above, there is evidence that IVH (in vivo) and Hb (in vitro) can induce CPEC cell death [27]. However, less is known about repair processes that might also occur. In the current study, Hb induced both proliferation and wound healing in CPECs primary cultures and Z310 cells. Multiple assays were used to confirm proliferation as each has limitation. Thus, for example, changes in the MTT assay could reflect changes in cell metabolic activity as well as cell number/viability. In experimental cerebral ischemia, Gillardon et al. found increased BrdU staining in the lateral, 3rd and 4th ventricle CPs [40] also indicating that brain injury can induce a proliferative response in the CP. The balance between CPEC death and repair will determine the effect of a brain injury, such as IVH, on overall CP size (which may then impact CSF secretion).

The mechanism by which Hb impacts CPEC cell proliferation is still unclear. However, the current study found an upregulation of mRNA levels of a number of growth factors (BDNF, IGF2, TGF-β1, and NGF) in isolated CPs exposed to Hb. This suggests a potential autocrine effect. Nilsson et al. found that IGF2 (the most highly expressed growth factor in the current study) caused proliferation of CPECs [41]. The current study also suggests that the CP may be an important source of growth factors after IVH. There is evidence of CP-induced neuroprotection via growth factor release in a number of injury and disease states, including stroke and traumatic brain injury [42,43]. Previous studies have shown that Hb and iron cause ventricle dilation [9] and the involvement of TGF-β1 and -2 in hydrocephalus formation [10,11]. While CP-derived growth factors may be important in CP repair [41], they may also have other effects on the ventricular system and CSF outflow sites.

We found in this study that Hb induced CPEC cell proliferation, but recent studies reported that exposing human primary CP cells to CSF from preterm infants with IVH or freshly prepared fetal Hb or MetHb showed no alteration [44] or decreased cell viability [27]. For example, Fejes et al. [44] reported no difference in human CPEC viability after exposure to CSF from human IVH patients. This contradiction could result from differences in Hb sources/forms (commercial vs. freshly prepared), as well as cell type and sensitivity (rat vs. human). In the case of preterm infants with IVH, the CSF could also contain other factors that influence the cells, like cytokines, growth factors, or products of hemorrhage (e.g., blood breakdown products). These factors could exert a toxic effect on the cells, overshadowing any positive effects of Hb on cell proliferation. To fully understand this contradiction, it would be important to compare the specifics of each study, especially the exact experimental conditions (e.g., Hb type/source, CSF source, and exposure concentration and duration) and the cells used. Comparing results from animal cells and human cells [44] could also provide more understanding of the role of Hb in PHH.

A recent study has shown CSF hypersecretion by the CPECs in IVH, which was linked to upregulation of Na/K/Cl cotransporter (NKCC1) activity in CPECs [16]. In this study, we found that Hb upregulated NKCC1 mRNA expression in isolated CPs. However, while there is evidence that NKCC1 is involved in CSF secretion [21,22], a recent study by Sadegh et al. has suggested that NKCC1 can be involved in CSF reabsorption and can reduce hydrocephalus [23]. NKCC1 is a bidirectional transporter with its net effect dependent on intra- and extracellular ion gradients.

Hb also upregulated expression of claudin-2 by the CP. Unlike ‘barrier-forming’ claudins, such as claudin-5, claudin-2 can form pores in tight junctions, which have been postulated to be involved in CSF secretion [45]. Our findings in this study are thus consistent with and support previous studies and indicate possible hypersecretion of CSF by CPECs in IVH. These changes might aid in the clearance of intraventricular blood, but it might also contribute to PHH.

Traditionally, PHH is assumed to result from a block in CSF flow pathways or fibrosis at CSF outflow sites rather than secretion. In this study, we found Hb induced growth factor-mediated CP proliferation and upregulation of CSF secretion-related proteins. These findings, along with previous reports, suggest that oversecretion CSF in IVH may contribute to PHH and that alternative therapeutic targets may exist.

## 5. Conclusions

In conclusion, exposure of in vitro CP models to hemoglobin induces proliferation, wound repair, growth factor upregulation, and induction of the Na/K/Cl cotransporter NKCC1 and claudin-2. These changes may have important impacts on CP function in the setting of IVH.

## Figures and Tables

**Figure 1 biomedicines-13-01366-f001:**
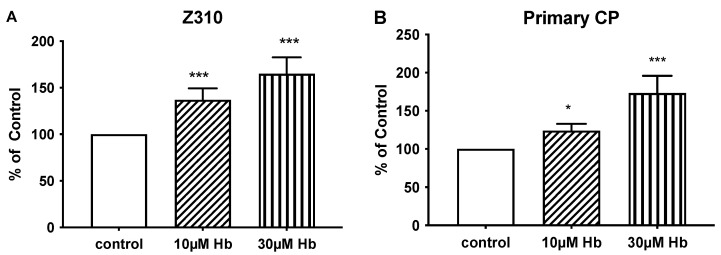
Impact of hemoglobin on choroid plexus epithelial cell proliferation. Comparison of cell proliferation induced by 24 h treatment with 10 or 30 µM Hb vs. control cultures of Z310 cells (**A**) and primary CPECs (**B**), as assessed by MTT assay. Values are means ± S.E., n = 5 independent experiments. * and *** indicate significant differences from controls at the *p* < 0.05 and *p* < 0.001 levels, respectively.

**Figure 2 biomedicines-13-01366-f002:**
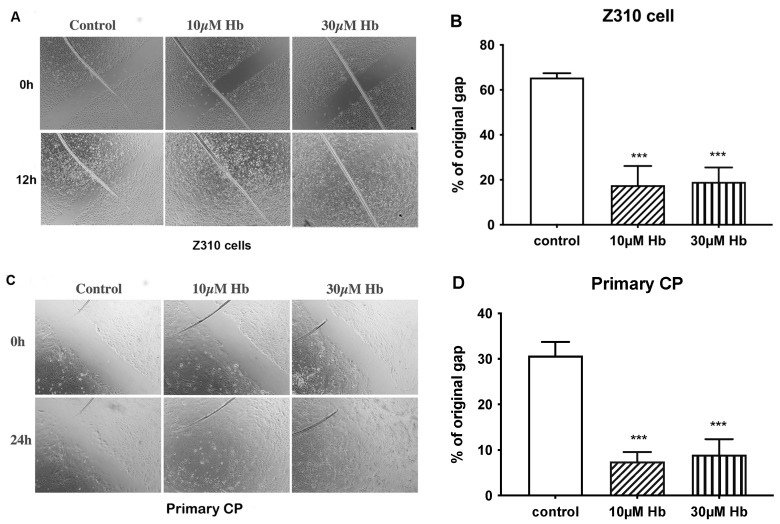
Impact of hemoglobin on choroid plexus epithelial cell healing. Scratch wound healing assay after treatment with 10 or 30 µM Hb vs. control in Z310 cells ((**A**,**B**), 12 h) and primary CPECs ((**C**,**D**), 24 h). Wound healing is assessed by measuring the gap width at the end of the experiment, expressed as a percentage of the original induced gap. Values are expressed as means ± S.E., n = 3 independent experiments. *** indicates significant differences from controls at the *p* < 0.001 level. Images were taken using a 10× objective lens.

**Figure 3 biomedicines-13-01366-f003:**
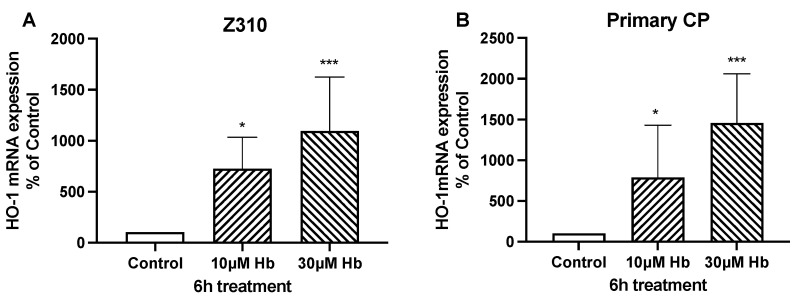
mRNA expression of HO-1 in Z310 cells (**A**) and primary CP cells (**B**) after 6 h of treatment with 10 µM or 30 µM Hb. Values are expressed as means ± S.E., n = 6–7 independent experiments. * and *** indicate significant differences from controls at the *p* < 0.05 and *p* < 0.001 levels, respectively.

**Figure 4 biomedicines-13-01366-f004:**
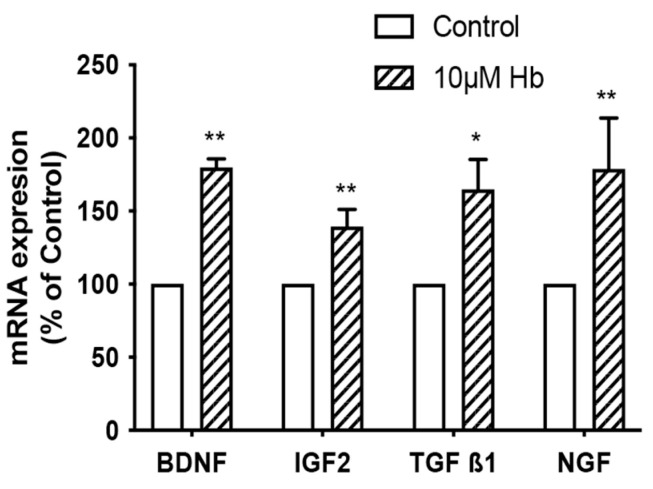
Impact of hemoglobin exposure on choroid plexus growth factor expression. mRNA expression of BDNF, IGF2, TGF-β1, and NGF in isolated rat CP tissue after 6 h of treatment with 10 µM Hb. Values are expressed as means ± S.E., n = 4–5 independent experiments. * and ** indicate significant differences from controls at the *p* < 0.05 and *p* < 0.01 levels, respectively. There were no significant differences in TGF-β2, TGF-β3, or bFGF2 mRNA levels (Table 2).

**Figure 5 biomedicines-13-01366-f005:**
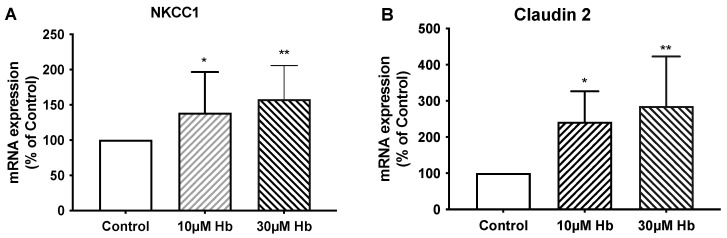
Effect of hemoglobin exposure on Na/K/Cl cotransporter (NKCC1) (**A**) and the tight junction protein claudin-2 (**B**) mRNA expression in isolated CP tissue after 6 h treatment with 10 µM or 30 µM Hb vs. control. Values are expressed as means ± S.E., n = 6–14 independent experiments. * and ** indicate significant differences from controls at the *p* < 0.05 and *p* < 0.01 levels.

**Figure 6 biomedicines-13-01366-f006:**
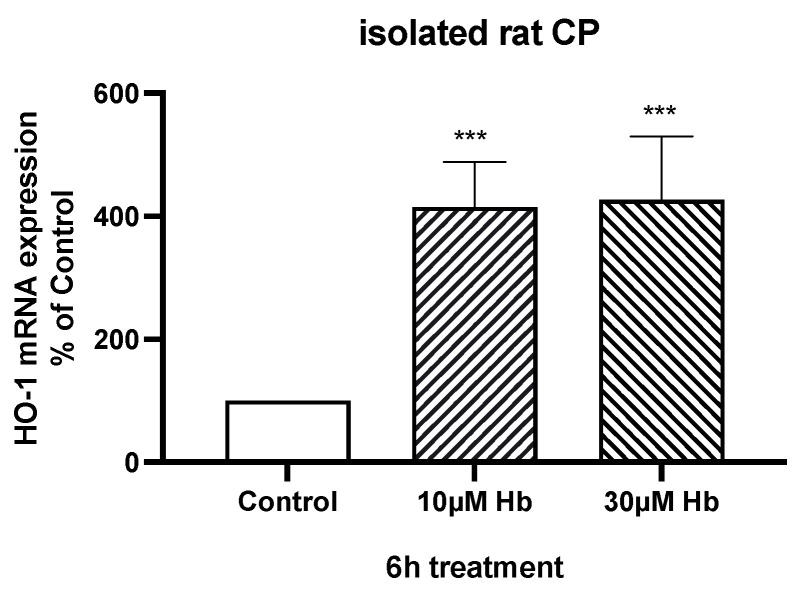
mRNA expression of HO-1 in isolated rat CP tissue after 6 h of treatment with 10 µM or 30 µM Hb. Values are expressed as means ± S.E., n = 5 independent experiments. *** indicates significant differences from controls at the *p* < 0.001 level.

**Table 1 biomedicines-13-01366-t001:** Primers used in qRT-PCR.

Gene	Accession No.	Forward (5′–3′)	Reverse (5′–3′)
*BDNF*	M61178.1	ATAATGTCTGACCCCAGTGCC	CTGAGGGAACCCGGTCTCAT
*IGF2*	NM_031511	TGTCTACCTCTCAGGCCGTACTT	TCCAGGTGTCGAATTTGAAGAA
*NGF*	NM_001277055.1	GAAACGGAGACTCCGTTCACC	GATTGTACCATGGGCCTGGA
*TGF-β1*	NM_021578.2	TGGCGTTACCTTGGTAACC	GGTGTTGAGCCCTTTCCAG
*TGF-β2*	AF153012.2	ATCGATGGCACCTCCACATATG	GCGAAGGCAGCAATTATGCTG
*TGF-β3*	NM_013174.2	AAGCGCACAGAGCAGAGAATC	AGTGTCAGTGACATCGAAG
*BFGF-2*	EF030430.1	AAGCGGCTCTACTGCAAG	CAGGCCCCGTTTTGGATCCG
*NKCC1*	AF051561	CCCGCGGCGCCCTCGTCT	GCACCGTGTCCCCGCCGTTCTG
*Na/K ATPase ∂1*	NM_012504.1	ATCTGCTCCGACAAGACTGGAACTCT	TTCTGGGGCGCCCTTCATCAC
*Na/K ATPase β1*	NM_013113.2	ACTGAAATTTCCTTCCGTCCTAAT	CGTCAGAGGGTAAGTCTCCAA
*Na/K ATPase β2*	U45946.1	TCCTGGGCGATATTATGAGCAACC	CCACGCGGGCAGCAAACTT
*Claudin-2*	NM_001106846.2	AGGGTTTCCGGGACAATAA	TAAAGTATCTGGTAGGGTTGCC
*Claudin-11*	BC070927.1	CGGGCTGGATCGGTGCTGTG	AATGGAACGCCCGAGGAAAGGAG
*Occludin*	AB016425.1	GCGACCGCGGTGGAGTTG	AAGCCGCTGCCGTAAGGGTAGT
*HO-1*	BC091164.1	AGGTCAAGCACAGGGTGACAGA	CTAGCAGGCCTCTGGCGAAGA
*GAPDH*	AF106860.2	AGACAGCCGCATCTTCTTGT	CTTGCCGTGGGTAGAGTCAT

**Table 2 biomedicines-13-01366-t002:** Control mRNA expression of (A) growth factors and (B) ion transporters and tight junction (TJ) proteins in isolated choroid plexus tissue, and the impact of 6 h hemoglobin (Hb; 10 μM) exposure.

**(A) Growth Factors**		
**mRNA**	**Control Expression** **(Ratio to GAPDH)**	**Hb-Induced Expression** **(% of Control)**
BDNF	0.00072 ± 0.00012	180 ± 6 **
IGF-2	4.3 ± 0.21	139 ± 12 **
TGF-β1	0.038 ± 0.0044	158 ± 25 *
NGF	0.095 ± 0.016	179 ± 35 **
BFGF-2	0.014 ± 0.003	153 ± 41
TGF-β2	0.17 ± 0.05	99 ± 35
TGF-β3	0.021 ± 0.003	90 ± 14
**(B) Ion Transporters and TJ Proteins**		
**mRNA**	**Control Expression** **(Ratio to GAPDH)**	**Hb-Induced Expression** **(% of Control)**
NKCC1	0.0008 ± 0.0001	150 ± 15 **
Na/K ATPase ∂1	1.43 ± 0.12	90 ± 2.3
Na/K ATPase β1	1.57 ± 0.16	97 ± 9.2
Na/K ATPase β2	0.35 ± 0.07	117 ± 21
Claudin-2	0.017 ± 0.008	254 ± 53 *
Claudin-11	0.0007 ± 0.0003	546 ± 268
Occludin	0.25 ± 0.0027	91 ± 18

Values are expressed means ± SE; n = 3–5. * and ** indicate significant differences from control at the *p* < 0.05 and 0.01 levels, respectively.

## Data Availability

The original contributions presented in this study are included in the article. Further inquiries can be directed to the corresponding author.

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
