# Peer review of "Growth Factors and the Choroid Plexus: Their Role in Posthemorrhagic Hydrocephalus"

_biomedicines, 2025, doi:10.3390/biomedicines13061366_

Round 1

Reviewer 1 Report

Comments and Suggestions for Authors

In this original article entitled “Growth factors and the choroid plexus: Role in post-hemorrhagic hydrocephalus” Ye H et al. aimed to investigate the impact of hemoglobin (Hb) on the choroid plexuses (CP) under in vitro conditions. For this purpose, primary CP epithelial cells isolated from rat CPs, Z310 cells and rat CP tissue as well.

Importantly, intraventricular hemorrhage (IVH) and subarachnoid hemorrhage (SAH) are severe, frequently seen complications in premature infants and in adults, respectively, therefore biomarker research in this field is still a ‘hot-topic’.

Overall, the manuscript has some merits and fits to the scope of the Journal. Despite of these interesting findings, some improvements should be addressed before final decision.

Major comments:

  1. One of the major criticisms is that the Authors used 10-30uM Hb for treatments, however it is recommended to clarify which Hb form was used (e.g. oxyHb, MetHb, ferrylHb or heme?). In addition, these treatments significantly increased cell growth in case of both cell types (Fig.1), however in the literature recent studies showed no alteration or decreased cell viability after such treatments (PMIDs: 25441622, 32210955, 33573677 ). How could you interpret these controversial results? Please clarify it in the Discussion involving more relevant publications.
  2. Another limitation is that in this manuscript the Authors only presented the CP tissue mRNA results, however, cell proliferation and healing were investigated in CP epithelial cells and Z310 cells. It is recommended to measure mRNA expression in all used cell types.
  3. Did the Authors measure HMOX1 or other different inflammatory mRNAs (e.g. IL1B, IL6) as positive controls?
  4. In Table 2. part B among the mRNA result which Hb concentration is showed (10 or 30uM)?
  5. Sentences on page 7, lines 181-186 are difficult to understand, please clearly revised this paragraph. In addition, NKCC1 and Claudin 2 mRNA results should be presented only 1 figure as Fig. 4. showing the individual results on the bar graphs. In Table 2. Claudin-11 also showed a marked increase, however, this alteration did not explain in the text.
  6. The Authors used rat CP epithelial cells, Z310 cells and rat CP tissue for in vitro experiments, however it is recommended to compare the results in the Discussion with more human studies which used human CP epithelial cells (HCPEpiC) (PMIDs: 25441622, 34445350) to understand the increased risk of mortality and morbidity after IVH or SAH.

Minor comments:

  1. Abstract should be revised with more details, highlighting the main results.
  2. English language of the manuscript needs to be revised and improved. Several spelling errors have been found throughout the manuscript.
  3. It is recommended to use higher resolution and equal size of Figures.
  4. Use the same Font size and type in the Tables and Figures according to the Journal recommendations.

Comments on the Quality of English Language

English language of the manuscript needs to be revised and improved. Several spelling errors have been found throughout the manuscript. 

Author Response

We sincerely thank the reviewers for their detailed and insightful comments, which has helped us improve the quality and clarity of our manuscript.

Reviewer 1:

  1. One of the major criticisms is that the Authors used 10-30uM Hb for treatments, however it is recommended to clarify which Hb form was used (e.g. oxyHb, MetHb, ferrylHb or heme?). In addition, these treatments significantly increased cell growth in case of both cell types (Fig.1), however in the literature recent studies showed no alteration or decreased cell viability after such treatments (PMIDs: 25441622, 32210955, 33573677 ). How could you interpret these controversial results? Please clarify it in the Discussion involving more relevant publications.

The Hb was purchased from BP Biomedicals (cat# 151234). Initially, it is a mixture of met-and oxyhemoglobin but incubations were performed in a 20% oxygen atmosphere. The Hb was dissolved in culture medium without FBS and diluted to desired concentrations.

We have also addressed the difference between our results and recent reports by other researchers in Discussion and cited relevant publications.

  1. Another limitation is that in this manuscript the Authors only presented the CP tissue mRNA results, however, cell proliferation and healing were investigated in CP epithelial cells and Z310 cells. It is recommended to measure mRNA expression in all used cell types.

The reason we chose CP tissue to measure mRNA expression is that original tissue is often considered better than primary cultured cells or cell lines because it most accurately represents the in vivo physiological state of the tissue, maintaining the complex cellular interactions and heterogeneity that can be lost when cells are isolated and cultured, even in a primary cell culture, making it more relevant for studying tissue function and response to treatments; while cell lines, being immortalized, often undergo genetic alterations that significantly change their behavior compared to the original tissue. We will expand our measurement to Z310 and primary CP cells in future study.

  1. Did the Authors measure HMOX1 or other different inflammatory mRNAs (e.g. IL1B, IL6) as positive controls?

Yes, We did measure HMOXI (HO-1) mRNA expression in Z310 cells, primary cultured rat cp cells and isolated rat cp and found that 24h treatment of Hb (10 and 30µM) significantly upregulated HO-1 mRNA expression in the cells and tissue. Those results were not presented in this manuscription.

  1. In Table 2. part B among the mRNA result which Hb concentration is showed (10 or 30uM)?

It is 10µM, which is included in Table 2’s legend.

  1. Sentences on page 7, lines 181-186 are difficult to understand, please clearly revised this paragraph. In addition, NKCC1 and Claudin 2 mRNA results should be presented only 1 figure as Fig. 4. showing the individual results on the bar graphs. In Table 2. Claudin-11 also showed a marked increase, however, this alteration did not explain in the text.

We have combined fig 4 and fig 5 as a new fig 4.

  1. The Authors used rat CP epithelial cells, Z310 cells and rat CP tissue for in vitro experiments, however it is recommended to compare the results in the Discussion with more human studies which used human CP epithelial cells (HCPEpiC) (PMIDs: 25441622, 34445350) to understand the increased risk of mortality and morbidity after IVH or SAH.

We have addressed this in Discussion and cited relevant publications.

Minor comments:

  1. Abstract should be revised with more details, highlighting the main results.
  2. English language of the manuscript needs to be revised and improved. Several spelling errors have been found throughout the manuscript. We have checked spelling and corrected typos.
  3. It is recommended to use higher resolution and equal size of Figures. We have increased the figures’ resolution.
  4. Use the same Font size and type in the Tables and Figures according to the Journal recommendations. We have done that. 

Reviewer 2 Report

Comments and Suggestions for Authors

The study by Ye et al., investigates the response of choroid plexus epithelial cells to hemoglobin (Hb), which is particularly relevant in the context of intracerebral or subarachnoid hemorrhage, conditions often associated with intraventricular hemorrhage. In such cases, Hb is present in ventricular cerebrospinal fluid (CSF) and comes into direct contact with the choroid plexus structures of the ventricles. However, the impact of Hb exposure on choroid plexus epithelial cells remains unclear. In this study, the authors use cultured cells (immortalized and primary cultures) and ChP explants to interrogate the response of ChP to Hb, particularly with regards to proliferation and transcription of key growth factor, ion channel, and tight junction genes. While this study presents an interesting area of research, there are several important aspects that require further clarification, and a more cautious interpretation of the results is warranted.

Major concerns: 

  1. Hemoglobin Preparation and Control Treatment: the method of preparing Hb and the control treatments used are not clearly described. Without this information it is challenging to assess whether the experimental design is valid. For example, how was the Hb sourced? What were the control conditions (e.g. PBS, vehicle)? These details are cruicial for understanding the specificity and reliability of the observed effects.
  2. Interpretation of the MTT Assay: The primary assays used to conclude that Hb increased ChPe proliferation are the MTT and scratch assays (see 3 for an alternative interpretation of the scratch assay that should be considered). The MTT assay measures metabolic activity and cell viability. Therefore, the increased signal detected in the MTT assay after Hb treatment could be due to a variety of factors, including but not limited to increased proliferation of the cells. Alternatively, the increased signal could result from increased metabolic activity, ROS production or altered redox state following Hb treatment. What is the control? Free Hb itself can directly reduce MTT, which would lead to high absorbance values only in the Hb conditions. The methods describes that the cells were washed with PBS prior to the assay, however, any residual Hb in the treatment groups could result in this artifact that needs to be properly controlled. Additional methods to quantify cell proliferation, such as Ki67 immunostaining or EdU labeling following Hb treatment is necessary to rule out these alternative explanations.
  3. Clarification of the Scratch Wound Healing Assay: Details of the scratch would healing assay are unclear. Why is there a scrape, moving from the top left to the bottom right of the image? Is this the "scratch", or is the clear smudge lacking cells that goes from the top right to the bottom left of the image what was quantified? If it is the smudge, which is what was initially assumed prior to reading the methods that describe a 28G scrape, then this needs to be clarified. Also, the size of the representative images for 30um Hb are smaller than for the controls and 10um, but it isn't clear why that is. Further, the interpretation appears to be that the reduced scratch in the Hb treatments is due to proliferation, however, it could also be enhanced migration after Hb exposure. Again, methods to determine proliferative index of the cells under control and Hb conditions are necessary to support this conclusion.
  4. In vitro Limitations and Potential for Exacerbating PHH: While the hypothesis that Hb-induced upregulation of growth factors and CSF secretion-related genes may contribute to PHH is intriguing, the limitation of the work being entirely in vitro must be highlighted. Further, the suggestion of alternative therapeutic targets for PHH is promising, especially is Hb is implicated in promoting proliferation and CSF secretion. However, the interpretation needs to account for the potential downside of these effects, such as if excessive growth factor signaling or changes in CSF dynamics might lead to compensatory overproduction of CSF or even exacerbate hydrocephalus. Additionally, there could be a fine line between therapeutic promotion of healing and uncontrolled growth, especially if growth factors like TGFb have fibrotic effects.
  5. Distinguishing Regenerative and Toxic Effects of Hb: It is important to distinguish between the potential regenerative effects of Hb and its toxic effects (e.g. oxidative stress, iron dysregulation, etc). Assessing ROS generation or cell apoptosis would help clarify whether the observed effects are truly beneficial or a response to damage. 

Overall, while the study provides valuable insight into the potential effects of Hb on ChPe cells, several important methodological and interpretative issues need to be addressed. Further clarification of the hemoglobin preparation, control treatments, and experimental details is needed. Additionally, the authors should carefully consider the in vitro limitations and the potential implications of their findings for PHH, acknowledging both the regenerative and toxic potential of Hb exposure. Finally, more comprehensive experiments to measure proliferation and assess stress (ROS, apoptosis, etc) will help to better define the effects of Hb on ChPe cells.

Author Response

Reviewer 2: 

  1. Hemoglobin Preparation and Control Treatment: the method of preparing Hb and the control treatments used are not clearly described. Without this information it is challenging to assess whether the experimental design is valid. For example, how was the Hb sourced? What were the control conditions (e.g. PBS, vehicle)? These details are crucial for understanding the specificity and reliability of the observed effects.

We have addressed these in the revision.

  1. Interpretation of the MTT Assay: The primary assays used to conclude that Hb increased ChPe proliferation are the MTT and scratch assays (see 3 for an alternative interpretation of the scratch assay that should be considered). The MTT assay measures metabolic activity and cell viability. Therefore, the increased signal detected in the MTT assay after Hb treatment could be due to a variety of factors, including but not limited to increased proliferation of the cells. Alternatively, the increased signal could result from increased metabolic activity, ROS production or altered redox state following Hb treatment. What is the control? Free Hb itself can directly reduce MTT, which would lead to high absorbance values only in the Hb conditions. The methods describes that the cells were washed with PBS prior to the assay, however, any residual Hb in the treatment groups could result in this artifact that needs to be properly controlled. Additional methods to quantify cell proliferation, such as Ki67 immunostaining or EdU labeling following Hb treatment is necessary to rule out these alternative explanations.

We have added cell counting and total cell protein measurement to support MTT assay results.

  1. Clarification of the Scratch Wound Healing Assay: Details of the scratch would healing assay are unclear. Why is there a scrape, moving from the top left to the bottom right of the image? Is this the "scratch", or is the clear smudge lacking cells that goes from the top right to the bottom left of the image what was quantified? If it is the smudge, which is what was initially assumed prior to reading the methods that describe a 28G scrape, then this needs to be clarified. Also, the size of the representative images for 30um Hb are smaller than for the controls and 10um, but it isn't clear why that is. Further, the interpretation appears to be that the reduced scratch in the Hb treatments is due to proliferation, however, it could also be enhanced migration after Hb exposure. Again, methods to determine proliferative index of the cells under control and Hb conditions are necessary to support this conclusion.

The scrape is a maker (induced by scratching the plate surface with the needle tip of an insulin syringe) for taking images.

  1. In vitro Limitations and Potential for Exacerbating PHH: While the hypothesis that Hb-induced upregulation of growth factors and CSF secretion-related genes may contribute to PHH is intriguing, the limitation of the work being entirely in vitro must be highlighted. Further, the suggestion of alternative therapeutic targets for PHH is promising, especially is Hb is implicated in promoting proliferation and CSF secretion. However, the interpretation needs to account for the potential downside of these effects, such as if excessive growth factor signaling or changes in CSF dynamics might lead to compensatory overproduction of CSF or even exacerbate hydrocephalus. Additionally, there could be a fine line between therapeutic promotion of healing and uncontrolled growth, especially if growth factors like TGFb have fibrotic effects.

Yes, in vivo studies need to be done in the future to confirm our in vitro results.

  1. Distinguishing Regenerative and Toxic Effects of Hb: It is important to distinguish between the potential regenerative effects of Hb and its toxic effects (e.g. oxidative stress, iron dysregulation, etc). Assessing ROS generation or cell apoptosis would help clarify whether the observed effects are truly beneficial or a response to damage. 

We are planning to do this in the short future.

Round 2

Reviewer 1 Report

Comments and Suggestions for Authors

In the revised manuscript, entitled “Growth factors and the choroid plexus: Role in post-hemorrhagic hydrocephalus” by Ye H et al. the authors have just partly improved the quality of the paper and some of the responses are remained incomplete. As Biomedicines is a Q1 Journal, the reviewer could not accept it for publication in present form, and major modifications are still needed before final decision.

The Hb was purchased from BP Biomedicals, however this company is not found.

Abstract should be revised with more details, highlighting the main results.

The Reviewer did not detect plagiarism, however some paragraphs showed significant similarity mainly in the Methods and Discussion, thus it is recommended to carefully rephrase the manuscript.

Sentences on page 8, lines 186-190 are remained difficult to understand, please clearly revised this paragraph. Results in the text and Figure 4B do not match. In addition, the manuscript should be presented in an intelligible fashion and written in standard English. English language of the manuscript needs to be improved; it should be revised by an English native speaker. Several typographical and grammatical errors have been found throughout the manuscript (e.g. “in the this study”, “uesed”, “Hb’s role”, “sencitivity”, etc.).

HMOX1 mRNA results should be also presented in the manuscript.

There is a new paragraph in the Results section on page 5 (lines 144-147), however the exact test protocols (methodology) are missing from the Methods, and it is not mentioned in the Discussion. Revise it with more details.

New references should be also drafted based on the Journal recommendations (e.g. Ref. 44, etc.).

Comments on the Quality of English Language

Listed among the comments.

Author Response

We sincerely thank the reviewers again for their detailed and insightful comments, which has helped us improve the quality and clarity of our manuscript.

Reviewer 1

Comments and Suggestions for Authors

In the revised manuscript, entitled “Growth factors and the choroid plexus: Role in post-hemorrhagic hydrocephalus” by Ye H et al. the authors have just partly improved the quality of the paper and some of the responses are remained incomplete. As Biomedicines is a Q1 Journal, the reviewer could not accept it for publication in present form, and major modifications are still needed before final decision.

Abstract should be revised with more details, highlighting the main results.

We have added more details of our main results in the Abstract.

The Reviewer did not detect plagiarism, however some paragraphs showed significant similarity mainly in the Methods and Discussion, thus it is recommended to carefully rephrase the manuscript.

We did check and tried to reduce similarity in our manuscript.

Sentences on page 8, lines 186-190 are remained difficult to understand, please clearly revised this paragraph. Results in the text and Figure 4B do not match. In addition, the manuscript should be presented in an intelligible fashion and written in standard English. English language of the manuscript needs to be improved; it should be revised by an English native speaker. Several typographical and grammatical errors have been found throughout the manuscript (e.g. “in the this study”, “uesed”, “Hb’s role”, “sencitivity”, etc.).

These typos have been corrected.

HMOX1 mRNA results should be also presented in the manuscript.

HMOXL (HO-1) mRNA results were presented in the manuscript.

There is a new paragraph in the Results section on page 5 (lines 144-147), however the exact test protocols (methodology) are missing from the Methods, and it is not mentioned in the Discussion. Revise it with more details.

The related protocols were added to the Methods.

New references should be also drafted based on the Journal recommendations (e.g. Ref. 44, etc.).

We have done that in the revision.

Reviewer 2 Report

Comments and Suggestions for Authors

Inclusion of experimental details (Hb sourcing and preparation) has improved the manuscript. However, while the authors performed a few additional experiments those data do not sufficiently address previous concerns and almost none of the previous comments were addressed in the response. In particular, the major claim of this study is that Hb enhances proliferation of choroid plexus epithelial cells. An Hb only condition must be used to control for the reducing effect of Hb itself. Inclusion of cell counts and total protein is a start, yet the details of these new experiments were not provided. How were the cells prepared and harvested prior to these assays? These assays should also be performed on the primary CP cells to support the cell line data. Finally, as recommended before, proliferation assays measuring incorporation of a nucleotide analog or proportions of cells that express a proliferative marker (e.g. Ki67) is necessary to support the claim that Hb increases proliferation of CP cells. 

It is still unclear what the scratch would healing assay is measuring. Why is there etching (scrape) and wider cell-free region moving in the opposite orientation in the images? Also, the size of the representative images for 30um Hb are smaller than for the controls and 10um, but it isn't clear why that is. Further, the interpretation appears to be that the reduced scratch in the Hb treatments is due to proliferation, however, it could also be enhanced migration after Hb exposure. Again, methods to determine proliferative index of the cells under control and Hb conditions are necessary to support this conclusion (cell counts and protein levels do not suffice).

There are also many spelling errors in the added text, including "anount" (line 147), "conmercial" (line 240), "sencitivity" (line 241), and "uesed" (line 246).

Author Response

We sincerely thank the reviewers again for their detailed and insightful comments, which has helped us improve the quality and clarity of our manuscript.

Reviewer 2

Comments and Suggestions for Authors

Inclusion of experimental details (Hb sourcing and preparation) has improved the manuscript. However, while the authors performed a few additional experiments those data do not sufficiently address previous concerns and almost none of the previous comments were addressed in the response. In particular, the major claim of this study is that Hb enhances proliferation of choroid plexus epithelial cells. An Hb only condition must be used to control for the reducing effect of Hb itself. Inclusion of cell counts and total protein is a start, yet the details of these new experiments were not provided. How were the cells prepared and harvested prior to these assays? These assays should also be performed on the primary CP cells to support the cell line data. Finally, as recommended before, proliferation assays measuring incorporation of a nucleotide analog or proportions of cells that express a proliferative marker (e.g. Ki67) is necessary to support the claim that Hb increases proliferation of CP cells. 

We have tried western blots of Ki67 and PCNA to examine proliferation in Z310 cells. Those cells express very high level of PCNA under control conditions and we could not identify any increase after Hb treatment. The Ki67 western blots suggested an increase with Hb treatment, but were of insufficient quality for publication. Therefore, we rely on our three other measures of cell proliferation.

It is still unclear what the scratch would healing assay is measuring. Why is there etching (scrape) and wider cell-free region moving in the opposite orientation in the images? Also, the size of the representative images for 30um Hb are smaller than for the controls and 10um, but it isn't clear why that is. Further, the interpretation appears to be that the reduced scratch in the Hb treatments is due to proliferation, however, it could also be enhanced migration after Hb exposure. Again, methods to determine proliferative index of the cells under control and Hb conditions are necessary to support this conclusion (cell counts and protein levels do not suffice).

The additional scrape was for area identification for repeat measurements. All images were taken under same magnification.

There are also many spelling errors in the added text, including "anount" (line 147), "conmercial" (line 240), "sencitivity" (line 241), and "uesed" (line 246).

These typos have been corrected.

Round 3

Reviewer 1 Report

Comments and Suggestions for Authors

In the revised manuscript, entitled “Growth factors and the choroid plexus: Role in post-hemorrhagic hydrocephalus” by Ye H et al. the authors have improved the quality of the paper. Some new paragraphs and 2 Figures have been also added. The revised version is more consistent and better drafted, thus it is potentially acceptable for publication in Biomedicines.

As most of the important questions have been addressed in the revised manuscript, I have no further questions to raise, only minor comments:

1. Methods are missing from the Abstract.

2. Reagent manufacturers are missing from the Methods on page 3 (MTT-assay).

3. Fig.4A and Fig.4B are indicated in the text instead of Fig.5A and Fig.5B on page 8, lines 210 and 213.

Author Response

As most of the important questions have been addressed in the revised manuscript, I have no further questions to raise, only minor comments:

  1. Methods are missing from the Abstract.

Methods have been added to the Abstract.

  1. Reagent manufacturers are missing from the Methods on page 3 (MTT-assay).

MTT Reagent manufacturer has been added.

  1. Fig.4A and Fig.4B are indicated in the text instead of Fig.5A and Fig.5B on page 8, lines 210 and 213.

They have been corrected.

Thanks a lot.

Reviewer 2 Report

Comments and Suggestions for Authors

The response to previous comments are satisfactory.

Author Response

The response to previous comments are satisfactory.

Thanks a lot.